# High-Throughput Quantitation of Cannabinoids by Liquid Chromatography Triple-Quadrupole Mass Spectrometry

**DOI:** 10.3390/molecules27030742

**Published:** 2022-01-24

**Authors:** Jonathan Tran, Aaron C. Elkins, German C. Spangenberg, Simone J. Rochfort

**Affiliations:** 1Agriculture Victoria Research, AgriBio Centre, AgriBio, Bundoora, VIC 3083, Australia; jonathan.tran@agriculture.vic.gov.au (J.T.); German.Spangenberg@agriculture.vic.gov.au (G.C.S.); Simone.Rochfort@agriculture.vic.gov.au (S.J.R.); 2School of Applied Systems Biology, La Trobe University, Bundoora, VIC 3083, Australia

**Keywords:** cannabis, cannabinoids, LC-QQQ-MS, quantitation, high-throughput quantitation, analysis

## Abstract

The high-throughput quantitation of cannabinoids is important for the cannabis industry. As medicinal products increase, and research into compounds that have pharmacological benefits increase, and the need to quantitate more than just the main cannabinoids becomes more important. This study aims to provide a rapid, high-throughput method for cannabinoid quantitation using a liquid chromatography triple-quadrupole mass spectrometer (LC-QQQ-MS) with an ultraviolet diode array detector (UV-DAD) for 16 cannabinoids: CBDVA, CBDV, CBDA, CBGA, CBG, CBD, THCV, THCVA, CBN, CBNA, THC, Δ8-THC, CBL, CBC, THCA-A and CBCA. Linearity, limit of detection (LOD), limit of quantitation (LOQ), accuracy, precision, recovery and matrix effect were all evaluated. The validated method was used to determine the cannabinoid concentration of four different *Cannabis sativa* strains and a low THC strain, all of which have different cannabinoid profiles. All cannabinoids eluted within five minutes with a total analysis time of eight minutes, including column re-equilibration. This was twice as fast as published LC-QQQ-MS methods mentioned in the literature, whilst also covering a wide range of cannabinoid compounds.

## 1. Introduction

The use of cannabis has gained popularity as a therapeutic alternative in recent times, with many countries around the world legalising its use for medicinal purposes. In 2016, Australia legalized the use of cannabis for medicinal use, with the potential of legalisation for recreational use under investigation. As a medicinal agent, it is essential to identify the cannabinoid compounds present to understand the mechanisms of action. Currently, over 550 compounds have been identified in cannabis with 144 different cannabinoids isolated from cannabis [1]. The ever-growing cannabis industry has resulted in the development of more cultivars, all with different chemical compositions. Therefore, a need for a rapid, accurate and high-throughput method for chemical analysis, that can be used to determine the concentration of as many components as possible, is essential.

The main pharmacological compounds currently of interest are cannabidiol (CBD) and delta-9-tetrahydrocannabinol (THC), the two major phytocannabinoids present in cannabis buds, depending on the cultivar. CBD, a low psychoactive compound, has been proven to have beneficial health outcomes for epileptic individuals that suffer from Dravet syndrome, Lennox-Gastaut syndrome or tuberous sclerosis complex [2,3]. THC, although highly psychoactive, has been used as an analgesic for individuals suffering from multiple sclerosis, and as a treatment for urinary incontinence and spasticity symptoms [4]. Some of the conditions best suited to cannabis use as a medicinal agent exclusively effect children, such as Dravet syndrome. As such, the use of a cannabis resin, or oil, with no or negligible amounts of THC and elevated levels of CBD would be the ideal vehicle for administration.

While CBD, THC and their precursors, cannabidiolic acid (CBDA) and tetrahydrocannabinolic acid (THCA-A), respectively, are the major cannabinoids present in most cultivars, the minor cannabinoids have also been shown to have pharmacological benefits for various medical conditions [4,5]. Cannabinol (CBN), cannabigerol (CBG) and cannabichromene (CBC) have been shown to exhibit anti-inflammatory properties for various medical conditions, including irritable bowel syndrome [6,7,8,9,10]. There is also evidence to suggest that cannabigerolic acid (CBGA) and cannabidivarin (CBDV) have anti-convulsant properties and they may be useful in the treatment of epilepsy in conjunction with CBD [10,11]. Additionally, preliminary animal studies have shown that tetrahydrocannabivarin (THCV) may be useful in delaying the onset of neurodegenerative disorders, such as Parkinson’s disease [12]. Other cannabinoids that may be of interest include delta-8-tetrahydrocannabinol (Δ8-THC), cannabinolic acid (CBNA), cannabichromenic acid (CBCA), tetrahydrocannabivarin acid (THCVA), cannabidivarin acid (CBDVA) and cannabicyclol (CBL), which are generally found in trace amounts in cannabis; therefore, there is little research on the clinical effects of these compounds.

To ensure high-throughput sample analysis and processing, a rapid method to quantify the cannabinoids present in individual samples is necessary. One potential issue is the structural similarity of some cannabinoids. For example, Δ8-THC, is a structural isomer and derivative of Δ9-THC, which has an identical molecular weight, similar fragmentation ions and retention time, with the only difference being a shift in the position of the double bond [13]. Due to the higher abundance of Δ9-THC in some cultivars, rapid methods of analysis may result in co-elution, making it difficult to accurately determine the concentration of Δ8-THC. Additionally, the concentration of Δ8-THC tends to be in low abundance, such that quantitation software may include Δ9-THC peaks in Δ8-THC peaks. Many cannabinoids that have been discovered; however, the majority of them lack fully validated analytical methods for quantitation, and most studies at present validate methods for the major cannabinoid compounds [14]. Accurate quantitation of all the compounds present in a sample is essential in understanding the pharmacological benefits of using cannabis extracts containing multiple cannabinoids as a medicine; however, there are limitations, as not all isolated cannabinoids are commercially available. The quantitation of the 16 commercially available cannabinoids validated in this study are the main focus of many current studies in the literature.

Current methods for the quantitation of cannabinoids and secondary metabolites use either gas chromatography (GC) or liquid chromatography (LC), in tandem with a mass spectrometer (MS) and/or a diode array detector (DAD). High-performance liquid chromatography with a diode array detector (HPLC–DAD) has been used to quantify cannabinoid compounds in both cannabis oils and cannabis plant material [15], but this lacks the selectivity to distinguish between structurally similar, coeluting compounds. GC and GCMS is a more specific method of cannabinoid analysis, where all cannabinoid analytes are eluted in less than 20 min [14], with more recent methods optimised to reduce the analysis time to 7 min [16]. GCMS is used to quantify both terpenes and cannabinoids, but the limitation is that the operating temperature of the injection port may decarboxylate all acidic cannabinoids into their neutral derivatives [17].

LC is the alternate method of analysis for volatiles compounds as they can remain intact, and preparation steps are often easier, as derivatisation is not required [18]. Elkins et al. developed a validated method for 6 cannabinoids (CBD, THC, CBDA, THCA-A, CBN and CBC) in 20 min that was suitable for Good Manufacturing Practice environments and adhering with the Food and Drug Administration and the International Council for Harmonisation test methods using a UHPLC–DAD. The method allowed good separation but lacked a mass spectrometer that would have provided superior identification and selectivity [19]. Liquid chromatography triple quadrupole mass spectroscopy (LC-QQQ-MS), targeting specific cannabinoids, was used by Grauwiler et al. to detect four cannabinoid compounds—CBD, CBN, THC and THCA-A—in human plasma with all compounds eluting in 25 min [20]. This was recently improved by McRae et al., where the authors used a LC-QQQ-MS to quantitate 17 cannabinoids in 15 min [21]. QQQ-MS is effective for targeted analysis due to its selectivity but is unable to identify untargeted compounds [18].

Supercritical fluid chromatography (SFC) has been used due to its ‘green nature’, where the mobile phase is carbon dioxide rather than methanol or other organic solvents [22]. SFC can reduce the use of alcohol-related solvents, whilst still being able to achieve adequate separation, comparable to HPLC [14,21]. Wang et al. employed SFC to separate 11 cannabinoids in less than 10 min [22]. In addition, carbon dioxide supercritical fluid extraction is an effective method of extracting cannabis resin (primarily cannabinoids) from cannabis buds [21]. It does this by pressurising the carbon dioxide until it reaches supercritical conditions and is then able to extract cannabis resin whilst the plant matter is left behind, allowing for a cannabinoid-rich product for analysis. This process has limitations, since highly polar solutes are insoluble in carbon dioxide supercritical fluids.

Nuclear magnetic resonance (NMR) is often used for the identification of novel cannabinoids but can also be used for quantitation. Hazekamp et al. was able to quantify THCA-A, THC, CBD, CBDA and CBN in plant extracts and semi-pure, cannabinoid rich fractions within five minutes [23]. NMR is rapid, non-destructive and can characterise different regioisomers, but lacks sensitivity compared with mass spectrometry [24].

While there are many methods available to identify and quantify different cannabinoid compounds, many are either too slow (15–25 min), or not cost effective for high-throughput research. Our method uses an LC-QQQ-MS method, where we accurately quantitate 16 cannabinoids with a total analysis time of less than 8 min. This is a significant improvement on current industry standard methods for high-throughput screening, reducing overall analysis time while increasing the number of compounds analysed and still achieving high resolution and baseline separation.

## 2. Methods

### 2.1. Reagents and Standards

All reagents, water with 0.1% formic acid (mobile phase A), acetonitrile with 0.1% formic acid (mobile phase B), and methanol were HPLC grade and obtained from Fisher Scientific (Fair Lawn, NJ, USA). Primary standards for cannabidiolic acid (CBDA) tetrahydrocannabinolic acid (THCA-A) in acetonitrile, cannabidivarin acid (CBDVA), cannabidivarin (CBDV), cannabigerolic acid (CBGA), cannabigerol (CBG), cannabidiol (CBD), tetrahydrocannabivarin (THCV), tetrahydrocannabivarin acid (THCVA), cannabinol (CBN), cannabinolic acid (CBNA), tetrahydrocannabinol (THC), delta-8-tetrahydrocannabinol (Δ8-THC), cannabicyclol (CBL), cannabichromene (CBC), and cannabichromenic acid (CBCA) in methanol, at 1000 µg/mL, were commercially purchased from Novachem Pty Ltd. (Heidelberg West, Australia) as distributor for Cerilliant Corporation (Round Rock, TX, USA). Purity of standards were >98% according to the individual certificate of analysis.

In total, 2 separate mixed standards were prepared: 1 working standard at 100 µg/mL CBDA, CBD, CBN, THC, CBC and THCA-A in methanol; and the other 100 µg/mL CBDVA, CBDV, CBGA, CBG, THCV, THCVA, CBNA, Δ8-THC, CBL and CBCA in methanol. A 100 µg/mL standard was required for UV analysis; therefore, 2 separate standard preparations from the individual 1000 µg/mL neat stock were required. The working standard concentrations were 0.001, 0.01, 0.1, 0.25, 0.5, 1, 2.5, 5, 10, 50 and 100 µg/mL prepared as serial dilutions from their respective 100 µg/mL working standard. All standards were stored at −80 °C until required for analysis.

### 2.2. Sample Preparation and Extraction

Briefly, dried and ground cannabis inflorescences were obtained from the Victorian Government Medicinal Cannabis Cultivation Facility. Samples used were CannBio-2 (CB2), CannBio-3 (CB3), CannBio-4 (CB4), CannBio-5 (CB5) and an unnamed low THC strain. Samples were placed in liquid nitrogen for 1 min and ground to a fine powder using a SPEX SamplePrep 2010 Geno/Grinder for 1 min at 1500 rpm. This sample preparation and extraction was performed as described in a paper by Elkins et al. [19]. After grinding, 10 mg of each sample was weighed into 2 mL Axygen microtube and extracted with 1 mL of methanol, vortexed for 30 s, sonicated for 5 min and centrifuged at 13,000 rpm for 5 min. The supernatant was transferred to a 2 mL amber HPLC vial and diluted 1 in 10 to ensure responses were within the quantitative range of the instrument.

### 2.3. Pre-Extraction Spike Preparation

Recovery samples were prepared by spiking 10 mg of sample with 100 µL of 100 µg/mL standard and then prepared to 1 mL in methanol. A 1:10 dilution of the extract was performed to achieve a concentration of 1 µg/mL. The high spike (HS) was prepared by adding 500 µL of 100 µg/mL standard and followed the steps mentioned previously to achieve concentrations of 5 µg/mL. Samples were sonicated, vortexed and transferred into 2 mL amber HPLC vials as previously described. A further 1 in 2 dilution was required to ensure all cannabinoids fit within the calibration curve, making the total dilution 1 in 20. Final spikes concentrations were 0.5 µg/mL (low spike, LS); and 2.5 µg/mL (high spike, HS).

### 2.4. Post-Extraction Spike Preparation

To determine matrix effect, samples were extracted with methanol as outlined in Section 2.2, 100 µL of the extract was transferred to a 2 mL amber HPLC vial, spiked with 10 µL of 100 µg/mL standard and made to a final volume of 1 mL with methanol to achieve a concentration of 1 µg/mL. The HS used 50 µL of 100 µg/mL standard and was prepared as described to achieve a concentration 5 µg/mL. A further 1 in 2 dilution was required to ensure all cannabinoids fit within the calibration curve, making the total dilution 1 in 20. Final spike concentrations were 0.5 µg/mL (LS) and 2.5 µg/mL (HS).

### 2.5. Instrumentation Parameters

Analysis was performed on an Agilent Triple Quadrupole Mass Spectrometer 6460 coupled with an Agilent high-performance liquid chromatography 1290 Infinity II LC System equipped with a degasser, binary pump, temperature controlled autosampler, column compartment and UV-DAD. Agilent Mass Hunter Data Acquisition Version 10 was used for instrument control. The column used was a Phenomenex Luna Omega C_18_ 150 × 2.1 mm × 1.6 µm column with an injection volume of 5 µL. The mobile phases consisted of (A) water with 0.1% formic acid and (B) acetonitrile with 0.1% formic acid. Separation was achieved using the following gradient parameters: 0–5 min, 70% B; 5–7 min; 100% B, 7–7.1 min 70% B. This was followed by equilibration to initial conditions at a flow rate of 0.4 mL/min. The total runtime was less than 8 min with all compounds eluted in less than 5 min. The autosampler was maintained at 15 °C with the column temperature maintained at 40 °C. The UV-DAD was set up to acquire spectra at wavelengths of 280 nm and 214 nm. The QQQ-MS parameters were as follows: the gas temperature was set at 300 °C; the gas flow at 5 L/min; the nebuliser pressure at 45 psi; the sheath gas temperature at 250 °C; the sheath gas flow at 11 L/min. The ion spray voltage was at 3500 V at the capillary and 500 V at the nozzle.

### 2.6. Data Processing

Limit of detection (LOD) and limit of quantitation (LOQ) were determined using the LINEST function of excel and data from the Agilent Mass Hunter Quantitative Analysis where a signal ratio of 3.3:1 from baseline was used for LOD and LOQ was determined using signal ratio of 10:1 from baseline. R^2^ values and equations were calculated using Agilent Mass Hunter Quantitative Analysis software, where the calibration curve fit origins were forced through zero.

## 3. Results and Discussion

### 3.1. Method Validation

Method validation was evaluated for the following parameters: linearity, LOD, LOQ, accuracy, precision, matrix effect (ME) and recovery (RE). This was carried out in accordance with guidelines detailed by Peters et al. [25].

### 3.2. Compound Separation

The analysis was performed using LC-QQQ-MS. The precursor, quantifier, qualifier ions and collision energies are detailed in Table 1, with all compounds eluted within 8 min. Baseline separation for all cannabinoids was achieved except for compounds which exhibited the same retention time and molecular weight, which were Δ8-THC, Δ9-THC CBC and CBL, but these could be characterised and quantified using different product ions (Figure 1). Refer to Appendix A for MRM of scans of individual cannabinoids, including quantifier and qualifier ions.

### 3.3. Linearity, LOD and LOQ

The calibration curves consisted of 11 working standards prepared in methanol. Limit of detection (LOD) and limit of quantitation (LOQ) were determined to be approximately 0.1 µg/mL and from 0.08 to 0.71 µg/mL, respectively. The R^2^ values for each cannabinoid were 0.990 or better (Table 2), this is equivalent to the correlation coefficient criteria of McRae et al. [26], and is achieved whilst improving runtime to 8 min, compared with McRae’s 21 min total runtime [26]. Due to the high abundance of CBDA and THCA-A in samples UV-DAD was used for quantitation which is accurate up to 250 µg/mL according to Elkins et al. [19].

### 3.4. Accuracy and Precision

Accuracy and precision of the method was assessed by calculating the mean result of seven injections and determining the percent relative standard deviation (%RSD) of the repeat injections. This was performed on both standards and seven independent extracts of each CannBio strain. Repeated injections of the 0.25, 1 and 5 µg/mL standards resulted in a %RSD < 2.0 for all analytes that are within the linear range of the method (Table 3). Mean cannabinoid content and %RSD for each CannBio strain was determined with values ranging between 2 and 6% (Table 4). The precision values obtained was comparable to the values obtained by McRae et al. [26] (1.4 to 6.1%); however, our method was able to achieve these results in less than half the time. THCA was found to be highly abundant across all four strains, with CBDA also prominent in even ratio strains (CB2 and CB3), as expected. CBGA, CBG, THCVA, CBNA, THC and CBCA were found in relatively low abundance in all four strains with CBGA, THC and CBCA showing to be the most prominent. CBDVA, CBD and CBC were only observed in CB2 and CB3, indicating a pathway link to CBDA. There were no cannabinoids exclusive to the high THC strains (CB4 and CB5), this is not surprising given no high CBD lines were available during the validation. CBDV, THCV, CBN and CBL were not observed in any samples. This is not unexpected, given the low concentration of the acidified variant and thermal decarboxylation required to facilitate the conversion.

### 3.5. Recovery and Matrix Effect

Working standards were used to spike samples for pre- and post-spikes to evaluate recovery and matrix effect at 0.5 and 2.5 µg/mL concentrations. Matrix effect was defined as ion suppression. Recovery (RE) and matrix effect (ME) was calculated using the following formula:
(1)$$\%\mathrm{ME}\;\mathrm{or}\;\%\mathrm{RE}=\frac{\mathrm{spiked}\;\mathrm{sample}-\mathrm{no}\;\mathrm{spike}\;\mathrm{sample}}{\mathrm{spike}\;\mathrm{level}}\times 100$$

Pre-extraction recovery values ranged from 73.0% to 126.2% across all cannabinoids at both spike concentrations in all CannBio samples (Table 5). The LS for CBD, THC and CBCA yielded higher than expected results, ranging from 131.4% to 158.1%, this was not observed for the HS and is likely due to high endogenous levels in the samples. CBDA and THCA-A were quantified by UV, with recovery values ranging from 71.2% to 101.0%. Recovery values for CBDA, determined by MS for CB4 and CB5, ranged from 105% to 108.8%. This confirms that at low levels the method is robust enough to accurately quantify samples with low concentrations of CBDA. Δ8-THC and THCA-A spiked on a separate low THC strain to determine method efficiency. Recovery values ranged between 93.9% and 115.5% (Table 6).

Post-extraction values ranged from 72.5% to 120.6% across all cannabinoids at both spike concentrations (Table 7). Again, the CBDA matrix effect was determined by MS for CB4 and CB5, with values ranging from 98.1 to 112.1%. CBDA and THCA-A were quantified using UV where values ranged from 85.9% to 102.4%. Δ8-THC and THCA-A was spiked in a separate low THC strain with post extraction values ranging from 96.7% to 115.2% (Table 8). When comparing both recovery and matrix effect values at for CBGA, CBG, THC and THCA compounds, there was a difference of 11.2%, 15.6%, 24.3% and 33.8%, respectively, compared with the calculated matrix effect value in the high spike of CB5, this is likely due to endogenous levels cannabinoid, within the sample, combined with the spike saturating the methanol, potentially resulting in an incomplete extraction and thus lower than expected recovery values. All other values were consistent between the pre- and post-spike samples.

The extraction efficiency of the method is very good, with most of the matrix effect and recovery samples within 10% RSD and returning values between 90 and 110%, indicative of full extraction (excluding outlier compounds mentioned previously). CBDV, CBDA and THCV are a few examples of a full extraction across all spikes in all four CannBio strains. Improvement to this process may be to have additional extractions, perhaps a method using 3 extractions of 500 µL of methanol, which will likely extract any leftover cannabinoid into the solution, resulting in a more efficient extraction at the detriment of time. However, the aim of this study was to make the entire process as rapid as possible, so these results are an acceptable compromise.

## 4. Conclusions

A rapid, high-throughput method for the quantitation of cannabinoids in neat cannabinoid standards was developed and fully validated for 16 compounds (CBDVA, CBDV, CBDA, CBGA, CBG, CBD, THCV, THCVA, CBN, CBNA, THC, Δ8-THC, CBL, CBC, THCA-A and CBCA) by LC-QQQ-MS. The method was optimised to achieve a rapid analysis time of eight minutes per sample, significantly improving current published methods. Linearity, limit of detection, limit of quantitation, accuracy, precision, spikes for matrix effect and recoveries have all been evaluated to give consistent results and acceptable RSD values. There is a limitation in the analysis of Δ8 and Δ9 THC, since these compounds co-elute due to their structural similarity and high endogenous levels of Δ9 compared with Δ8. This can be overcome by using longer methods of analysis or incorporating a multistep gradient; however, these changes would be at the cost of reduced throughput. Linearity, precision, accuracy, matrix effect and recoveries were assessed to be within acceptable limits. The LC-QQQ-MS in tandem with a UV-DAD is a rapid, cost-effective and ideal method for analysing major and minor cannabinoids in high-throughput commercial or research environments.

## Figures and Tables

**Figure 1 molecules-27-00742-f001:**
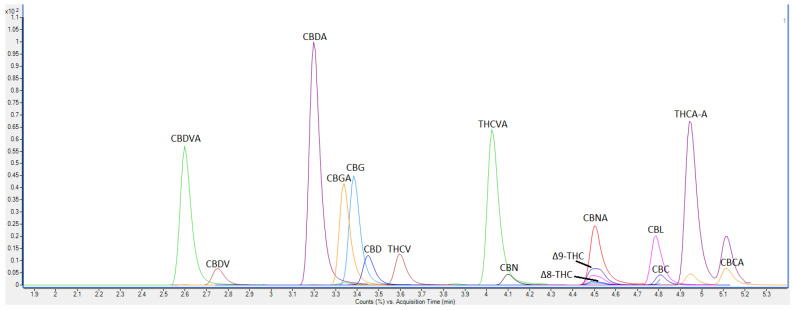
The dMRM elution profile of all 16 cannabinoids; cannabidivarin acid (CBDVA), cannabidivarin (CBDV), cannabidiolic acid (CBDA), cannabigerolic acid (CBGA), cannabigerol (CBG), cannabidiol (CBD), tetrahydrocannabivarin (THCV), tetrahydrocannabivarin acid (THCVA), cannabinol (CBN), cannabinolic acid (CBNA), tetrahydrocannabinol (THC), delta-8-tetrahydrocannabinol (Δ8-THC), cannabicyclol (CBL), cannabichromene (CBC), tetrahydrocannabinolic acid (THCA-A), and cannabichromenic acid (CBCA). Standards using the LC-QQQ-MS.

**Table 1 molecules-27-00742-t001:** Retention time, precursor, qualifier, quantifier ions and collision energy of each respective cannabinoid compound.

Compound	RT (min)	Precursor (*m*/*z*) [M + H]^+^	Quantifier (*m*/*z*)	CE (eV)	Qualifier (*m*/*z*)	CE (eV)
**CBDVA**	2.35	331.2	313.1	10	191	26
**CBDV**	2.49	287.2	123	18	165	35
**CBDA**	2.95	359.2	341	10	219	30
**CBGA**	3.08	361.2	219	15	343	22
**CBG**	3.15	317.3	193	10	123	35
**CBD**	3.24	315.2	193.1	18	123	30
**THCV**	3.35	287.2	123	30	165	18
**THCVA**	3.79	331.2	313.1	10	191	30
**CBN**	3.91	311.2	208	34	195.1	22
**CBNA**	4.23	355.2	337.1	10	235.1	30
**THC**	4.30	315.2	217.2	22	165.1	22
**Δ8-THC**	4.30	315.2	119	35	159.7	30
**CBL**	4.56	315.2	235.1	20	165	26
**CBC**	4.63	315.2	259	20	217	18
**THCA-A**	4.73	359.2	219	6	233	26
**CBCA**	4.90	359.2	219	15	233	22

**Table 2 molecules-27-00742-t002:** Linear concentration range, correlation coefficient, limit of detection and limit of quantitation for cannabinoid standards tested.

Compound	Concentration Range (µg/mL)	Equation	R^2^	LOD (µg/mL)	LOQ (µg/mL)
**CBDVA**	0.1–5	y = 14,085 × x	0.998	0.10	0.24
**CBDV**	0.1–5	y = 786 × x	0.999	0.05	0.10
**CBDA**	0.1–10	y = 23,687 × x	0.999	0.05	0.16
**CBDA ***	1–100	y = 12 × x	0.997	0.10	0.24
**CBGA**	0.1–5	y = 5237 × x	0.995	0.10	0.08
**CBG**	0.1–5	y = 2516 × x	0.999	0.10	0.20
**CBD**	0.25–10	y = 1841 × x	0.995	0.12	0.35
**THCV**	0.25–2.5	y = 1103 × x	0.998	0.13	0.40
**THCVA**	0.1–2.5	y = 12,252 × x	0.999	0.10	0.25
**CBN**	0.1–10	y = 588 × x	0.998	0.10	0.27
**CBNA**	0.1–5	y = 8428 × x	0.991	0.10	0.25
**THC**	0.25–10	y = 182 × x	0.990	0.23	0.71
**Δ8-THC**	0.1–10	y = 88 × x	0.995	0.10	0.25
**CBL**	0.1–5	y = 3369 × x	0.999	0.10	0.11
**CBC**	0.025–10	y = 350 × x	0.999	0.12	0.35
**THCA-A**	0.1–10	y = 2079 × x	0.999	0.02	0.07
**THCA-A ***	1–100	y = 18 × x	0.998	0.12	0.37
**CBCA**	0.1–10	y = 1680 × x	0.990	0.10	0.16

For full compound names, refer to Figure 1. All equations were forced through zero. * Results calculated using UV-DAD set to 280 nm due to high endogenous levels in samples.

**Table 3 molecules-27-00742-t003:** %RSD values for 16 cannabinoid compounds in standard solution at 0.25, 1 and 5 µg/mL.

Compound	RT (min)	0.25 µg/mL	1 µg/mL	5 µg/mL
**CBDVA**	2.35	1.70	1.97	0.99
**CBDV**	2.49	1.96	2.06	2.00
**CBDA**	2.95	1.37	2.07	1.90
**CBDA ***	2.95	0.58	0.63	1.37
**CBGA**	3.08	2.07	1.99	1.36
**CBG**	3.15	1.83	1.68	0.71
**CBD**	3.24	2.65 **	1.71	1.33
**THCV**	3.35	5.07 **	2.09	1.51
**THCVA**	3.79	1.29	1.95	1.10
**CBN**	3.91	3.67 **	0.87	1.07
**CBNA**	4.23	1.81 **	1.78	0.78
**THC**	4.30	6.22 **	1.86	2.02
**Δ8-THC**	4.30	26.5 **	1.87	1.98
**CBL**	4.56	1.31	1.12	1.33
**CBC**	4.63	1.83 **	1.03	1.38
**THCA-A**	4.73	1.89	2.08	1.54
**THCA-A ***	4.73	0.82 **	0.86	1.47
**CBCA**	4.90	1.76	1.97	0.92

* Results calculated using UV-DAD set to 280 nm due to high endogenous levels in samples. ** Values marked are equal or below LOQ on the MS. RT—retention time (minutes).

**Table 4 molecules-27-00742-t004:** The mean concentrations and (%RSD) relative standard deviations of cannabinoid content for each CannBio sample. Concentration units (Conc.) are in µg/mL.

	CannBio-2		CannBio-3		CannBio-4		CannBio-5	
Compound	Conc.	%RSD	Conc.	%RSD	Conc.	%RSD	Conc.	%RSD
**CBDVA**	0.35	3.27	0.20	4.30	<LOQ	N/A	<LOQ	N/A
**CBDV**	<LOQ	N/A	<LOQ	N/A	<LOQ	N/A	<LOQ	N/A
**CBDA ***	77.2	4.30	101	2.53	<LOQ	N/A	<LOQ	N/A
**CBGA**	2.21	2.03	2.41	3.71	3.08	2.48	3.26	4.88
**CBG**	0.56	4.94	0.66	3.40	1.88	4.07	1.60	5.39
**CBD**	2.16	3.78	2.95	3.05	<LOQ	N/A	<LOQ	N/A
**THCV**	<LOQ	N/A	<LOQ	N/A	<LOQ	N/A	<LOQ	N/A
**THCVA**	0.34	3.70	0.19	3.86	0.91	3.51	0.90	5.92
**CBN**	<LOQ	N/A	<LOQ	N/A	<LOQ	N/A	<LOQ	N/A
**CBNA**	0.10	3.19	0.08	5.80	0.09	4.49	0.15	4.70
**THC**	2.36	4.22	3.06	3.44	5.62	4.23	5.42	4.94
**CBL**	<LOQ	N/A	<LOQ	N/A	<LOQ	N/A	<LOQ	N/A
**CBC**	0.20	4.43	0.30	2.34	<LOQ	N/A	<LOQ	N/A
**THCA-A ***	44.2	4.22	49.6	2.65	107	4.31	119	5.39
**CBCA**	5.23	3.94	5.75	2.46	5.18	2.99	5.34	3.58

* Results calculated using UV-DAD set to 280 nm due to high endogenous levels in samples.

**Table 5 molecules-27-00742-t005:** Pre-extraction spikes for recovery values in four different CannBio strains.

	CannBio-2		CannBio-3		CannBio-4		CannBio-5	
Compound	LS	HS	LS	HS	LS	HS	LS	HS
**CBDVA**	88.0	99.7	94.0	97.6	99.9	102.8	101.1	101.2
**CBDV**	92.6	95.2	94.2	93.0	96.9	92.8	96.2	89.7
**CBDA ***	ND	71.2	ND	91.5	84.8	91.8	88.5	90.4
**CBDA**	N/A	N/A	N/A	N/A	105.0	108.8	107.0	106.0
**CBGA**	81.7	89.1	88.7	78.8	106.4	81.5	103.7	67.9
**CBG**	88.4	85.1	97.5	82.0	103.5	78.3	98.8	76.8
**CBD**	153.7	95.0	158.1	92.0	106.4	104.4	110.8	99.4
**THCV**	96.5	91.4	91.6	89.5	94.3	91.7	95.1	90.1
**THCVA**	95.2	95.7	89.5	95.1	97.6	91.3	96.5	89.1
**CBN**	89.4	92.6	92.5	89.4	95.7	91.1	94.8	92.0
**CBNA**	90.1	93.0	77.2	89.2	76.1	ND	76.3	ND
**THC**	102.9	88.7	131.4	100.4	142.0	79.1	157.9	65.6
**CBL**	104.6	99.2	95.7	103.0	104.3	102.0	109.0	99.9
**CBC**	88.5	92.5	79.6	89.3	88.0	97.5	95.6	93.0
**THCA-A ***	ND	80.9	ND	101.0	ND	91.7	ND	68.6
**CBCA**	80.8	83.8	95.1	81.9	103.8	97.8	121.6	100.2

LS—low spike; HS—high spike. * Results calculated using UV-DAD set to 280 nm due to high endogenous levels in samples.

**Table 6 molecules-27-00742-t006:** Pre-extraction spikes for recovery for Δ8-THC and THCA-A in a low THC cannabis strain.

	Low THC Strain	
Compound	LS	HS
**Δ8-THC**	97.0	93.9
**THCA-A ***	115.5	99.8

LS—low spike; HS—high spike. * Results calculated using UV-DAD set to 280 nm due to high endogenous levels in samples.

**Table 7 molecules-27-00742-t007:** Post-extraction spikes for matrix effect in 4 different CannBio strains.

	CannBio-2		CannBio-3		CannBio-4		CannBio-5	
Compound	LS	HS	LS	HS	LS	HS	LS	HS
**CBDVA**	84.1	96.4	92.3	93.5	97.5	92.9	89.0	92.4
**CBDV**	87.8	102.7	84.2	98.3	92.2	92.6	86.6	95.0
**CBDA ***	ND	99.5	ND	107	100.8	96.0	85.9	93.3
**CBDA**	N/A	N/A	N/A	N/A	111.7	109.4	98.1	112.1
**CBGA**	73.4	88.7	82.8	77.7	125.9	89.7	99.8	79.1
**CBG**	78.6	95.6	88.9	92.5	123.1	93.7	100.3	92.4
**CBD**	100.9	91.6	102.1	82.3	85.6	98.1	83.1	102.9
**THCV**	91.4	103.3	94.6	99.1	99.8	99.8	92.4	96.1
**THCVA**	78.2	103.4	87.1	102.2	94	97.9	87.4	96.7
**CBN**	84.6	96.4	76.1	98.7	92.1	95.6	82.5	95.2
**CBNA**	112	91.7	98	85.7	85.4	70.6	90.8	79.6
**THC**	98.9	86.7	81.6	98.3	78.1	72.5	97.6	89.9
**CBL**	87.8	94.1	89.3	90.6	99.9	91.7	87.2	87.3
**CBC**	82.1	96.1	101.2	107.8	109.2	108.4	101.3	105.6
**THCA-A ***	ND	99.8	ND	100.2	ND	271	ND	102.4
**CBCA**	85.1	88.4	89.3	79.1	120.6	104.0	97.3	90.0

LS—low spike; HS—high spike. * Results calculated using UV-DAD set to 280 nm due to high endogenous levels in samples.

**Table 8 molecules-27-00742-t008:** Post-extraction spikes for matrix effect for Δ8-THC and THCA-A in a low THC cannabis strain.

	Low THC Strain	
Compound	LS	HS
**Δ8-THC**	115.2	104.6
**THCA-A ***	108.1	96.7

LS—low spike; HS—high spike. * Results calculated using UV-DAD set to 280 nm due to high endogenous levels in samples.

## Data Availability

Raw data available upon request.

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
