# Peer review of "High-Throughput Quantitation of Cannabinoids by Liquid Chromatography Triple-Quadrupole Mass Spectrometry"

_molecules, 2022, doi:10.3390/molecules27030742_

Round 1
Reviewer 1 Report
This submitted paper discusses an analytical method using liquid chromatography-tandem mass spectrometry (LC-MS/MS) for 16 different cannabinoids substances simultaneously.
Unfortunately, we could not find any novelty in this paper because many analytical methods for cannabinoids using LC-MS/MS have been published.
In addition, when analyzing many similar compounds at the same time, fewer than two Product ions are required, and the Product ions effective for quantification should be shown. In addition, this study did not use labeled cannabinoids compounds for quantification. In addition, this study does not use labeled cannabinoids compounds for quantification, which is not an effective analytical method because the analysis time is short and many compounds are not separated.
Therefore, the limit of detection (LOD), limit of quantitation (LOQ), accuracy, precision, recovery, etc. derived in this study are extremely unreliable.
For these reasons, we have judged that this paper is not suitable for publication in Molecules.
Author Response
Reviewer 1:
Thank you to the reviewer for your comments.
Unfortunately, we could not find any novelty in this paper because many analytical methods for cannabinoids using LC-MS/MS have been published.
The novelty of this paper is that improves on existing processes by reducing the analysis runtime to 8 minutes from 21 minutes whilst maintaining a correlation coefficient values >0.990, which are equivalent to McRae’s paper using the same method as outlined in line 216. In addition, we have maintained a high level of accuracy and precision, equivalent to or better than McRae’s paper at outlined in line 233 and 234.
In addition, when analyzing many similar compounds at the same time, fewer than two Product ions are required, and the Product ions effective for quantification should be shown.
The retention time, precursor masses and selection of two unique product ions for each compound, LC-QQQ can filter and distinguish our cannabinoid of interest and eliminate any risk of incorrect identification of compounds.
Two product ions quantifier and qualifier ensuring accurate identification, precursor ions and RT have been included in Table 1 as part of the original submission. A unique quantifier ion was chosen for each compound and used for all method validation assessments.
In addition, this study does not use labeled cannabinoids compounds for quantification, which is not an effective analytical method because the analysis time is short and many compounds are not separated.
Isotopically labelled cannabinoids were not included for this study due to Victorian Government DHS holding permits limiting quantities of schedule materials stored on site. Isotopically labelled standards could be used but would not improve the obtained results. The exception to this would be to use labelled d8-THC to enhance the spike recovery and matrix effect results due to the close structural similarity and precursor ion and product masses, all other compounds are either separated through the LC-QQQ’s ability to filter compounds by different product ion masses. CBL and CBC co-elute but once again, are separated using unique product ions for CBL and CBC separately.
Therefore, the limit of detection (LOD), limit of quantitation (LOQ), accuracy, precision, recovery, etc. derived in this study are extremely unreliable.
We disagree with the reviewer’s comment. The high quality data from the recovery and matrix effect values using unspiked and spiked samples with different endogenous cannabinoid profiles yield high quality validation data therefore the method is robust and reliable.
Reviewer 2 Report
Authors have developed a method for the "High-throughput quantitation of cannabinoids by liquid chromatography triple-quadrupole mass spectrometry". This is an interesting topic because improved method has been developed with the high throughput aim, and robust analytical methodologies. However, the manuscript has few suggestions that should be clarified before the manuscript can be considered for publication:
I have attached the PDF file. Please go through the comments and do the modifications.
- My main concern is the novelty of the manuscript that should be clarified. Please compare your extraction and chromatographic method with previously reported method in "result and discussion section". This will help to show the novelty of the study.
- Authors should clearly state the main novelty of the extraction procedure in relation to reported methods or you may clearly mention that you have adapted reported method.

Author Response
Reviewer 2:
Thanks to the reviewer for your comments.
1: My main concern is the novelty of the manuscript that should be clarified. Please compare your extraction and chromatographic method with previously reported method in "result and discussion section". This will help to show the novelty of the study.
Line 216, makes the comparison from our paper to a paper by McRae Et al, where our correlation coefficients are equivalent if not better whilst achieving a shorter analysis time of 8-minutes, compared to McRae et al. total runtime of 21-minutes. This is emphasised again in line 235. Additional writing has been added to emphasise the improvement this chromatographic method has over previous literature in Line 216.
2: Authors should clearly state the main novelty of the extraction procedure in relation to reported methods or you may clearly mention that you have adapted reported method.
Extraction procedure as was taken from a previous paper by Elkins et al., it has not been adapted or optimised.
Rephrase the sentence. Meaning is not clear
The sentence in line 27 has been changed from “As a medicinal agent it is essential to understand mechanisms of action and therefore the active constituents present” to “As a medicinal agent it is essential to identify cannabinoid compounds present to understand mechanisms of action” to make it clearer for the reader.
Please mention the purity of these standards.
Added “Purity of standards were >98% according to the individual certificate of analysis” to line 127 as requested.
Why you have prepared two separate mixture. Please mention clearly so that reader can understand clearly.
Added “A 100ug/ml standard was required for UV analysis therefore two separate standard preparations were necessary” to line 131, in response to the reviewers comments on the reason why two separate standard mixtures were prepared.
Is this the optimized method or adapted method? Please mention clearly
The extraction method was taken from a previous paper using the exact same method. Added “This sample preparation and extraction was performed as described as in Elkins et al.” to line 137.
Retention time is not given in table. Please provide the retention time.
Retention time column was added to Table 1.
Please check the formula for ME.
The formula is correct, however the formatting has been changed between line 256 and 257 to make the equation neater.
Reviewer 3 Report
This is a very well written text.
Add y-intercept with the slope to Table 2.
Author Response
Reviewer 3:
Thank you to the reviewer for your comments.
Add y-intercept with the slope to Table 2.
The equations in Table 2 were forced through zero, this is mentioned in line 184. Another footnote has been added to the table for clarity.
Round 2
Reviewer 1 Report
All the pointed out parts have been corrected and can be published.
This manuscript is a resubmission of an earlier submission. The following is a list of the peer review reports and author responses from that submission.
Round 1
Reviewer 1 Report
This manuscript (MS) written by Jonathan Tran et al. made a protocol for the quantification of 16 cannabinoids using a liquid chromatography triple-quadrupole mass spectrometer (LC-TQ-MS). The reviewer agrees with the general importance of the methodology development for the quantification of target compounds; however, this study is limited in the conducting of instrumental analysis using a general-purpose machine. In other words, this study seems to have little scientific progress. Accordingly, the reviewer feels this report is a better fit for more specialized journals in the field of methodology and/or should need an example for application in scientific fields.
Reviewer 2 Report
The manuscript titled: "High-throughput quantitation of cannabinoids by liquid chromatography triple-quadrupole mass spectrometry." submitted to Metabolites describing very interesting and important study related to high-throughput quantitation of cannabinoids. The study is very well designed and as authors mentioned proposed protocol improve time of analysis up to 16 cannabinoids namely: CBDVA, CBDV, CBDA, CBGA, CBG, 13
CBD, THCV, THCVA, CBN, CBNA, THC, Δ8-THC, CBL, CBC, THCA-A and CBCA.
Authors presented methodology describing instrumental parameters as well as sample preparation technique. Despite validation is done, I could not find requirements authors followed (FDA or other?)?
Additionally in Table 2 authors provided the results of LOD and concentration range., nevertheless LOD is much higher than lower point of concentration range, could you please explain?
Figure 1 presenting chromatogram, could you please provide full range of time from 0 up to 8 min? Do you have any eluting compounds in Void volume?
In table 3 authors presenting RSD of injections or RSD whole protocol including sample preparation?
Reviewer 3 Report
This paper discusses an analytical method using liquid chromatography-tandem mass spectrometry (LC-MS / MS) for 16 types of cannabinoids substances at the same time.
Many methods for analyzing cannabinoids using LC-MS / MS have been published, and unfortunately, the novelty of this paper was not clear.
Although the analysis time has been shortened, there is no point in reducing the time because many compounds have not been separated.
The MRM analysis in Fig. 1 should be described in Precursor ion and Production ion.
Also, when analyzing a large number of similar compounds at the same time, there are few production ions and two or more are required. Furthermore, the label cannabinoids compound is not used for quantification in this study. Therefore, the limit of detection (LOD), limit of quantitation (LOQ), accuracy, precision, recovery, etc. derived in this study are considered to be extremely unreliable.
For these reasons, I have determined that this paper is not suitable for publication in Metabolites.